# Advancing Esophageal Cancer Treatment: Immunotherapy in Neoadjuvant and Adjuvant Settings

**DOI:** 10.3390/cancers16020318

**Published:** 2024-01-11

**Authors:** Daniel Park, Won Jin Jeon, Chieh Yang, Dani Ran Castillo

**Affiliations:** 1University of California, San Francisco-Fresno Branch Campus, Fresno, CA 93701, USA; daniel.park2@tu.edu; 2Loma Linda University Medical Center, Loma Linda, CA 92354, USA; wjjeon@llu.edu; 3Department of Internal Medicine for UCSF, University of California, and UC Riverside, Riverside, CA 92521, USA; chiehy@medsch.ucr.edu; 4City of Hope-Duarte, Department of Hematology & Oncology, Duarte, CA 91010, USA

**Keywords:** locally advanced esophageal cancer, LAEC, neoadjuvant chemoradiation, immune checkpoint inhibitors (ICIs)

## Abstract

**Simple Summary:**

The management of locally advanced esophageal cancer poses considerable challenges and current strategies have significant risks. Moreover, despite various treatment regimens, response rates are inconsistent. In recent years, a new class of targeted therapy classified as ‘immunotherapy’ has changed the landscape of cancer management. The immune system is naturally designed to destroy malignant and aberrant cells; however, cancer cells have developed mechanisms to escape recognition. Immunotherapy, stimulates and enhances the patient’s own immune system, improving its ability to detect and eliminate cancerous cells more effectively. A thorough comprehension of how immunotherapy yields clinical benefit is intricately tied to the tumor microenvironment and its complex interactions across various pathways. Clinical trials have been conducted to determine the efficacy of immunotherapy in the management of esophageal cancer. This manuscript is aimed at understanding the evolving role of immunotherapy and esophageal cancer in both neoadjuvant and adjuvant settings to assist in the clinical decision making process.

**Abstract:**

Locally advanced esophageal cancer (LAEC) poses a significant and persistent challenge in terms of effective treatment. Traditionally, the primary strategy for managing LAEC has involved concurrent neoadjuvant chemoradiation followed by surgery. However, achieving a pathologic complete response (pCR) has proven to be inconsistent, and despite treatment, roughly half of patients experience locoregional recurrence or metastasis. Consequently, there has been a paradigm shift towards exploring the potential of immunotherapy in reshaping the landscape of LAEC management. Recent research has particularly focused on immune checkpoint inhibitors, investigating their application in both neoadjuvant and adjuvant settings. These inhibitors, designed to block specific proteins in immune cells, are meant to enhance the immune system’s ability to target and combat cancer cells. Emerging evidence from these studies suggests the possibility of a mortality benefit, indicating that immunotherapy may contribute to improved overall survival rates for individuals grappling with esophageal cancer. This manuscript aims to meticulously review the existing literature surrounding neoadjuvant and adjuvant immunotherapy in the context of LAEC management. The intention is to thoroughly examine the methodologies and findings of relevant studies, providing a comprehensive synthesis of the current understanding of the impact of immunotherapy on esophageal cancer.

## 1. Introduction

Esophageal cancer (EC) is the seventh most common cancer in the world [1]. In the United States in 2023, an estimated 21,560 new cases of EC were diagnosed, of which 16,120 resulted in death [2]. The two main histological types of EC are esophageal squamous cell carcinoma (ESCC) and esophageal adenocarcinoma (EAC). Each is a distinct entity in terms of clinicopathologic, histologic, and mutational characteristics. Globally, the majority of ECs are ESCC in histology, with the incidence of ESCC being on a downward trend [3]. Conversely, the incidence of EAC is rising, with studies pointing to gastroesophageal reflux disease (GERD) as a primary risk factor for EAC [4]. The pathogenesis of ESCC and EAC is illustrated in Figure 1.

Treatment of EC varies based on staging and patient characteristics. LAEC is often treated with curative intent, with neoadjuvant chemoradiation (nCRT) being shown to be effective in ESCC. Endoscopic submucosal dissection (ESSD) or surgery can be utilized in the early stages of EAC, and patients often undergo neoadjuvant CRT in addition to surgery [5]. The current standard treatment for unresectable locally advanced esophageal cancer is definitive chemoradiotherapy. Despite various studies investigating optimal chemotherapy regimens and alternative radiotherapy dose fractionation, the complete response rate remains low, and there continues to be a high rate of local recurrence [6]. EC exhibits elevated recurrence rates, with 35–50% of patients developing recurrence after curative intent esophagectomy within the first 12 months. Similarly, 30–50% of patients develop recurrence after CRT [7]. Immunotherapy, particularly the use of ICIs, represents a promising advancement in treatment options aimed at addressing the significant challenges associated with the high recurrence rates and poor prognosis in EC. This manuscript focuses on the role of immunotherapy in the neoadjuvant and adjuvant settings for LAEC. It identifies significant pathways in the tumor microenvironment (TME) and other biomarkers that advocate for the use of immunotherapy in EC, while also discussing novel targeted therapies.

## 2. Immunotherapy and the Tumor Microenvironment in EC

The tumor microenvironment (TME) is a heterogenous ecosystem of immune cells, endothelial cells, perivascular cells, fibroblasts, and extracellular matrices that is disrupted by cancer cells to facilitate proliferation and survival. Characteristics of the TME include chronic inflammation, acidosis, hypoxia, and immunosuppression. The upregulation of inflammatory signaling pathways is a critical component of EC survival. Interleukin-6 (IL-6) is a potent cytokine that binds to the transmembrane protein glycoprotein 130 and triggers downstream pathways such as Ras-MAPK, STAT1, STAT3, PI3K, and SHP2 [8]. The IL6/STAT3 pathway has been found to be upregulated in EC. The constitutive activation of STAT3 induces tumor cells to produce vascular endothelial growth factor (VEGF), IL-10, and inhibit dendritic cell maturation [9].

ESCC and EAC have unique mechanisms of pathogenesis. In ESCC, carcinogens such as tobacco smoke and alcohol induce chronic irritation and inflammation in the esophageal endothelium through reactive oxygen species (ROS) production as well as direct toxic effects. In EAC, gastric acid damages the esophageal endothelium and promotes ROS production. The direct insult activates the sonic hedgehog (SHH) pathway between the stroma and adjacent damaged epithelium, causing intestinal metaplasia [8]. Ultimately, through persistent DNA damage, neoplasia is induced. ROS activates pro-survival pathways such as phosphatidylinositol 3-kinase (PI3K), MAPK1/2, and NF-kB [8]. Figure 2 illustrates several aspects of the TME.

The activation of proinflammatory pathways and direct epithelial damage creates the toxic inflammatory environment for tumor cell development through blocking apoptosis, stimulating angiogenesis, and immune evasion. Immune evasion in malignant cells may occur through overexpression of PD-L1 on cancer cells. The PD-1/PD-L1 pathway inhibits and exhausts the host T cell response, allowing malignant cells to escape immune surveillance. Increased levels of PD-L1 have been detected in both ESCC and EAC [10,11]. Another mechanism of immune evasion occurs through CTLA-4, a transmembrane protein that is exclusively expressed on activated T cells which, when activated, prevents T cells from destroying cancer cells. The upregulation of CTLA-4 allows malignant cells to escape immune surveillance [12]. Given the diverse and multiple mechanisms by which cancer cells achieve a state of immortality, the introduction of immunotherapy has revolutionized cancer treatment. Novel therapeutics such as anti-PD1 agents (nivolumab, pembrolizumab, camrelizumab, tislelizumab, sintilimab and toripalimab), anti-PD-L1 agents (atezolizumab and avelumab), and anti-CTLA4 agents (ipilimumab and tremelimumab) have been utilized to reduce mortality in esophageal cancer.

## 3. The Relationship between Immunotherapy and Chemoradiation

The promising results of combination immunotherapy with chemoradiation in esophageal cancer further demonstrate the efficacy of combinatorial approaches in the first-line setting. There are several synergistic mechanisms between immunotherapy and chemotherapy/radiotherapy which have been identified by preclinical and molecular studies [13]. One of the main mechanisms is the reciprocity between immunotherapy and chemoradiation therapy in their ability to prime one another for esophageal cancer [14]. Radiotherapy reprograms the TME by promoting an inflammatory environment [15]. Specifically, radiotherapy triggers antitumor immunity, leading to the release of tumor antigens and an increase of circulating tumor-specific CD8+ T cells [15]. These changes in the TME can be favorable for subsequent treatment with immunotherapy. Studies have also demonstrated increased expression of PD-L1 after treatment with immunotherapy. This phenomenon is also seen with chemotherapy, and the overall increase in expression of PD-L1 can be directly targeted by immune-checkpoint inhibitors (ICIs) [16]. Radiotherapy has also been shown to lead to an abscopal effect. The direct and indirect antitumor effects of radiotherapy may be further augmented by immunotherapy [17]. Immunotherapy may serve as a primer for improved response to subsequent lines of chemoradiation. For example, recent studies have suggested that immunotherapy may lead to radiosensitization of the TME. Not only does radiotherapy lead to changes in the TME, but immunotherapy also leads to clinically significant alterations in the TME. Immunotherapy has been shown to increase CD8+ T cells, which can increase radiosensitivity and lead to increased tumor regression [14].

The theoretical rationale for the combination lies in the immunomodulatory effects of ionizing radiation (IR). IR can induce a robust antitumor immune response by influencing various stages of the cancer-immunity cycle, as opposed to selectively targeting specific steps, which is typical with ICIs. These effects include enhancing the release and presentation of tumor antigens, promoting the priming and activation of immune cells, increasing the density of tumor-infiltrating lymphocytes, facilitating the recognition of tumor cells by T cells, and amplifying the overall antitumor effect. In recent animal studies, combining CIRT (Carbon Ion Radiotherapy) with ICI treatment proved to be more effective in eliminating both locally irradiated tumors and distant, non-irradiated cancers. This suggests a synergistic anticancer impact when utilizing carbon ions in conjunction with immunotherapy [18].

Studies have also specifically explored the immunomodulatory effects of Stereotactic Ablative Radiotherapy (SBRT). Combining SBRT, immunotherapy, and low-dose RT has been shown to facilitate abscopal tumor regression. SBRT has been proven to activate cGAS-STING and NF-kB signaling. The activation of the cGAS-STING pathway results in the secretion of chemokines and cytokines, creation of a hypoxic environment, and triggering of the alternative and classical complement systems. These processes are critical in reprogramming the tumor microenvironment. Researchers have also observed an increase in the ratio of CD8+ to suppressor Treg cells and a decrease in CTLA-4 expression following RT [19].

## 4. Immunotherapy in the Neoadjuvant Setting

The advent of immunotherapy has created novel treatment regimens for EC. Landmark trials discussed in this manuscript demonstrated the role of immunotherapy in the adjuvant setting of EC; however, novel research has been conducted on immunotherapy and its role in the neoadjuvant setting. Initially, clinical trials evaluated immunotherapy in conjunction with chemoradiotherapy, and subsequent trials evaluated neoadjuvant chemoimmunotherapy as well as immunotherapy plus antiangiogenic therapy. Trials and studies that evaluated the safety and/or feasibility of neoadjuvant immunotherapy in EC include PALACE-1, PERFECT, SIN-INCE, Shen et al., Yang et al., and He et al. [20,21,22,23,24,25] Trials and studies that evaluated responses include ESONICT-1, PEN-ICE, SIN-ICE, NICE, He et al., and Liu et al. [26,27,28,29,30,31,32,33].

Trials to characterize the role of neoadjuvant immunotherapy in conjunction with chemoradiation include PALACE-1 and PERFECT. PALACE-1 analyzed preoperative pembrolizumab combined with chemoradiotherapy for esophageal squamous cell carcinoma. The trial included twenty resectable ESCC patients, and preoperative treatment included carboplatin, paclitaxel, radiotherapy, and pembrolizumab. Within 4–6 weeks after preoperative therapy, patients underwent surgery. Eighteen patients underwent surgery after preoperative pembrolizumab plus concurrent chemoradiotherapy, and the pathologic complete response was 55.6%, with the percentage of transcription factor 2 positive cells found to be significantly higher in specimens of the pathologic complete response group than the non-pathologic complete response group [21]. The PERFECT trial was unique in that it investigated neoadjuvant immunotherapy and EAC. The trial combined neoadjuvant chemoradiotherapy with atezolizumab in resectable EAC. Of the forty subjects enrolled in the study, thirty-four patients received all cycles of atezolizumab, and thirty-three underwent surgery. In total, 25% of patients achieved complete pathological remission. Overall, there was no statistically significant difference in response or survival between the trial cohort and neoadjuvant chemoradiation cohort [22].

The literature to date has shown a greater number of studies and trials regarding the role of neoadjuvant chemoimmunotherapy. ESONICT-1 was a single-center phase 2 trial that evaluated neoadjuvant sintilimab plus chemotherapy for locally advanced esophageal squamous cell carcinoma. Thirty patients were enrolled and received cisplatin, albumin-bound paclitaxel, and sintilimab. Within 4–6 weeks after treatment, patients underwent esophagectomy. The pathologic complete response rate of the primary tumor was 21.7%, and the major pathologic response rate of the primary tumor was 52.2%. From radiologic evaluations, the objective response rate was 67%, and the disease control rate was 97% [26]. ESONICT-2 analyzed toripalimab combined with docetaxel and cisplatin. Twenty patients completed two cycles of neoadjuvant therapy and underwent minimally invasive esophagectomy. The objective response rate was 70%, and the pathologic complete response rate of the primary tumor was 16.7% [29]. PEN-ICE evaluated neoadjuvant pembrolizumab and chemotherapy in resectable esophageal cancer. Eighteen patients underwent chemoimmunotherapy and thirteen patients progressed to surgery. Postoperative pathology revealed a major pathologic response rate of 69.2% and a pathologic complete response rate of 46.2% [27]. SIN-ICE evaluated sintilimab in combination with chemotherapy for neoadjuvant treatment of resectable esophageal cancer. Twenty-three patients were enrolled, and after chemoimmunotherapy seventeen patients underwent surgery, with sixteen receiving R0 resection and one receiving R1 resection. Among the patients, 76.5% showed a good pathologic response and downstaging. Pathologic complete response was achieved in 35.3% of cases and major pathologic response in 52.9% [23]. A pilot study by Yang et al. evaluated neoadjuvant camrelizumab plus chemotherapy in treating locally advanced esophageal squamous cell carcinoma patients. Following chemoimmunotherapy, all patients underwent surgery. The overall remission rate was 81.3%, and the disease control rate was 100%. The pathologic complete response rate was 31.5% [25]. Table 1 details further trials on neoadjuvant immunotherapy. The literature shows that there has been promising evidence of improved pathologic complete response with neoadjuvant immunotherapy regimens; however, more robust clinical trials need to be conducted to fully understand the therapeutic and adverse effect profile of these checkpoint inhibitors. Appendix A contains a list of ongoing neoadjuvant clinical trials.

## 5. Immunotherapy in the Adjuvant Setting

### 5.1. Adjuvant Immunotherapy and ESCC

The CROSS trial demonstrated that preoperative chemoradiation improved survival amongst patients with potentially curable esophageal or esophagogastric-junction cancer utilizing a regimen of carboplatin and paclitaxel and concurrent radiotherapy, followed by surgery. However, given the high recurrence rate of EC, further modalities for treatment have been investigated, such as immunotherapy in the adjuvant setting [35]. The phase III trial CheckMate 577 was conducted as no adjuvant treatment had been established for patients who are at a high risk for recurrence of esophageal or gastroesophageal junction cancer. Adults with resected (R0) stage II or III esophageal or gastroesophageal junction cancer who received neoadjuvant chemoradiotherapy received nivolumab at a dose of 240 mg/2 weeks for 16 weeks, followed by 480 mg/4 weeks, for 1 year. Of the patients in this study, 29% had ESCC. The benefit of nivolumab was found to be greatest with ESCC, with a disease-free survival (DFS) of 29.7 months, and after a median follow-up of two years, nivolumab was associated with a reduced risk of recurrence or death by 31% [36].

Within the ongoing phase III trials exploring immunotherapies for patients diagnosed with esophageal cancers (ECs) (Appendix A), the KEYSTONE-002 trial stands out as a multicenter, prospective, randomized-controlled study focusing on the neoadjuvant application of immunotherapy. This trial is structured into two groups: the experimental group, receiving pembrolizumab in conjunction with neoadjuvant chemoradiotherapy, and the control group, undergoing neoadjuvant chemoradiotherapy alone [37]. Appendix A lists the ongoing clinical trials for ESCC and EAC.

### 5.2. Adjuvant Immunotherapy and EAC

Recent trials have demonstrated the role of immunotherapy not only in ESCC but also in EAC. In the previously mentioned CheckMate 577 trial, 71% of the patients in the study had EAC (563/794 patients). Among patients who had EAC, the study showed improved DFS with those who received nivolumab compared to placebo (19.4 months vs. 11.1 months, HR 0.75, 95% CI 0.59–0.96). Of all the patients, 99% in both groups received prior neoadjuvant concurrent CRT [36]. Thus, nivolumab does play a role in immunotherapy in patients with EAC.

Clinical evidence suggests that immunotherapy with CRT may be an encouraging therapeutic option for patients with locally advanced ESCC; however, further research is necessary due to lack of long-term follow-ups from studies/trials [1].

## 6. Biomarkers of Prognostic Value

Despite the recent approvals of immune checkpoint inhibitors for EC, there is a need for further exploration of biomarkers of prognostic and predictive value. Recent trials have demonstrated the significance of PD-L1 expression in EC, encompassing both ESCC and EAC. In ESCC, KEYNOTE 590 demonstrated superior OS and PFS in patients who received pembrolizumab plus chemotherapy, compared to placebo plus chemotherapy. Notably, patients with PD-L1 CPS ≥ 10 demonstrated the most improved OS and PFS (HR 0.62, 95% CI 0.49–0.78, *p* < 0.0001 and 0.51, 95% CI 0.41–0.65, *p* < 0.0001, respectively). The improvement in OS was more pronounced in patients with ESCC who had CPS ≥ 10 (HR 0.57, 95% CI 0.43–0.75, *p* < 0.0001) [38]. Similarly, in patients with EAC, CheckMate 649 showed significant improvement in OS and PFS in patients with PD-L1 CPS ≥ 5 and ≥1 and even improved outcomes in patients with CPS score < 5; however, there was a lack of comparable OS in patients with PD-L1 CPS < 1 (HR 0.92, 95% CI 0.70–1.23, *p* = 0.2041) [39]. Interestingly, the CheckMate 577 trial found no significant improvement in DFS with nivolumab compared to placebo in patients with PD-L1 expression ≥1, but showed improved DFS after nivolumab in patients with PD-L1 expression <1 [36]. Overall, PD-L1 expression in tumor cells of both ESCC and EAC carries significant value in patient selection and management of EC.

In addition to PD-L1 expression and CPS, the genomic profiling of EC has led to advances in our understanding of other potential biomarkers and actionable mutations. Studies have demonstrated a diverse composition of the tumor microenvironment in ESCC and EAC. Next-generation sequencing (NGS) of EC demonstrated that EAC expressed significantly higher levels of HER2 overexpression compared to ESCC (*p* < 0.001) [40]. Furthermore, ESCC demonstrated higher PD-L1 expression compared to EAC. Recent clinical trials have shown mixed results with PD-L1 expression, with some demonstrating improved response with increased PD-L1 expression, and others showing similar responses to ICIs regardless of PD-L1 status [41].

Another area of interest for future study is that of the mutational profiles of EC, specifically the differences between ESCC and EAC. EAC has been shown to demonstrate statistically higher mutation rates of APC, ARID1A, KRAS, and SMAD4 [40]. These trends not only highlight the clinicopathologic differences between ESCC and EAC but also shed light on the role of such mutational and TME differences as prognostic and predictive biomarkers in EC.

## 7. Current Treatment Strategies

### ESMO 2023

In its 2023 guidance paper on the management of esophageal squamous cell carcinoma (ESCC), the European Society for Medical Oncology (ESMO) outlines a treatment algorithm that integrates immunotherapy [42]. The guidance paper emphasizes the addition of chemoradiotherapy (CRT) based on the CROSS trial, which demonstrated the benefits of incorporating carboplatin, paclitaxel, and radiation therapy. Notably, the CheckMate 577 trial is integrated, as adjuvant nivolumab improved disease-free survival in cases with R0 resections and residual pathological disease. Acknowledging that survival-focused strategies may not suit every patient, the standard of care for ESCC involves definitive CRT with up to 65 Gy, following the CROSS regimen [35]. For esophageal adenocarcinoma (EAC), neoadjuvant CRT and neoadjuvant chemotherapy using FLOT (5-fluorouracil, leucovorin, oxaliplatin, docetaxel) are recommended equally. Despite a complete clinical response to preoperative CRT or chemotherapy, patients with resectable disease are advised to undergo surgery, as limited data support a ‘wait and watch’ strategy [42]. The recommended treatment strategy is depicted in Figure 3.

## 8. Future Directions in the Management of EC

### 8.1. Next-Generation Checkpoint Inhibition

Beyond PD-1 and CTLA-4, other checkpoint inhibitory receptors have been utilized and targeted with immune checkpoint inhibitors, including T cell immunoreceptors with immunoglobulin and ITIM domain (TIGIT). TIGIT is usually activated by immune cells and binds to CD155 (PVR) and CD112 (PVRL2, nectin-2) [43]. CD155T/TIGIT signaling has been found to regulate immune response and promote tumor progression [44]. Similar to PD-1, high TIGIT expression is related to poor prognosis [45]. The current focus of clinical trials involving anti-TIGIT monoclonal antibodies is to combine with other traditional ICIs. The STAR-221 trial is a randomized phase III trial of anti-TIGIT therapy, domvanalimab, in addition to the standard combination of anti-PD-1 therapy and chemotherapy in patients with locally advanced unresectable or metastatic gastric, GEJ, and esophageal adenocarcinoma [46].

### 8.2. Tumor-Infiltrating Lymphocyte (TIL) Therapy

Tumor-infiltrating lymphocytes are considered a component of TMEs, and have been considered a possible prognosis predictor [47]. In a study by Wang et al., RNA-seq sequencing data were collected from 119 ESCC tumors and matched with adjacent normal samples. They found that the samples were heavily infiltrated by B cells and plasma cells compared to activated T cells. Furthermore, they also found several unexpected associations between tumor-infiltrating B cells and prognosis [48]. In another study, the clonal expansion of TIL was significantly induced within ESCCs after definitive CRT, and the ratio of PD-L1 mRNA to CD8B mRNA in TME was significantly associated with poor prognosis [49]. Adoptive cell therapy with tumor-infiltrating lymphocytes (TILs) has also been utilized in different solid tumors, including melanoma [50]. The application of TIL therapy in esophageal cancers could be one of the future directions of research. 

### 8.3. Personalized Treatment Strategies

Personalized treatment strategies with tumor profiling may offer a future direction for EC. In addition to the aforementioned PD-L1 status, this includes the status of human epidermal growth factor receptor 2 (HER2), microsatellite instability (MSI), and mutational burden [51]. In 2017, the Cancer Genome Atlas Research Network (TCGA) performed a comprehensive molecular analysis of 164 esophageal carcinomas based on somatic copy-number alterations, DNA methylation, mRNA, microRNA expression, and reverse-phase protein array data. Their analysis found that EACs were closer to gastric cancers compared to ESCCs, especially gastric CIN cancers [52]. A study by Kim et al. studied the endoscopic biopsy specimens of 64 EAC patients. Various statistical and informatical methods were applied to gene expression data to identify potential biomarkers of prognostic significance. Genes associated with protection or risk were linked to forecasting overall survival, where AKR1B10 and SOX21 were identified as protective genes, while DKK3 and SPP1 were categorized as risk genes [53].

### 8.4. CAR-T Cell Therapies

Chimeric antigen receptor T cell (CAR-T cell) therapy is a recently developed therapy targeting relapsed or refractory hematologic malignancies. Its potential clinical benefits may also extend to EC. EphA2 CAR-T cell immunotherapy designed to target EphA2 (erythropoietin-producing hepatocellular receptor A2) has been used in EphA2-positive glioma cells [54]. In cases of ESCC, EphA2 overexpression is associated with a negative prognosis [55]. Consequently, EphA2 CAR-T cell immunotherapy has been considered for use for ESCC. In 2018, EphA2 CAR-T cells showed a dose-dependent cell killing effect and significantly elevated TNF-α and IFN-γ levels in ESCC cells in vitro [54]. HER2 overexpression has also been found in esophageal cancers, which has led to the development of CAR-T cells targeting the HER2 antigen. HER2 CAR-T cells were found to successfully kill HER2-positive tumor cells in vitro and suppress established esophageal squamous cell carcinoma cells in a subcutaneous xenograft BALB/c nude mouse model in vivo [56]. There are other developing CAR-T therapies, such as MUC1-CAR-T cells and CD276-specific CAR-T cells, showing some benefits both in vitro and in vivo in ECs [57,58]. Currently, these CAR-T therapies have not been employed clinically in EC, yet they have shown promise in preclinical studies using both in vitro and in vivo models. Further research and evidence could endorse the utilization of CAR-T therapies in treating ECs.

### 8.5. Cancer Vaccines

Cancer vaccines are a promising new topic, designed by stimulating T cells targeting specific tumor-associated antigens (TAAs) to kill tumor cells [41]. Kageyama et al. conducted a trial involving twenty-five patients diagnosed with advanced/metastatic esophageal cancer. The clinical study focused on the NY-ESO-1 protein vaccine and revealed dose-dependent effects of the CHP-NY-ESO-1 vaccine on overall survival, with no reports of severe adverse events [59]. Additionally, in a trial by Iinuma et al., eleven patients participated in a phase I trial involving the combination of multiple-peptide vaccine therapy and CCRT for individuals with unresectable ESCC. Their findings indicated that the combined treatment was well-tolerated, yielding six cases of complete response [60].

## 9. Conclusions

The management of EC is complex and nuanced, marked by a poor 5-year overall survival rate with existing treatment modalities. Moreover, despite treatment measures, EC is prone to recurrence/metastasis. Research has shed light on the tumor microenvironment, immunotherapy’s synergistic effects with chemoradiation, and crucial prognostic biomarkers. Immunotherapy has demonstrated through novel trials and studies that it may significantly affect clinical outcomes in EC. Further research efforts may solidify its role in the neoadjuvant and adjuvant settings.

## Figures and Tables

**Figure 1 cancers-16-00318-f001:**
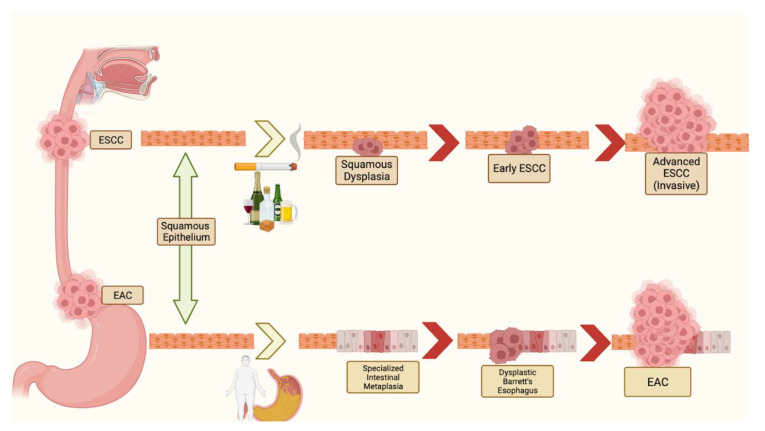
Pathogenesis of ESCC and EAC. Major risk factors for ESCC include tobacco and alcohol use. Major risk factors for EAC include GERD and obesity. For ESCC after esophageal injury from smoking or alcohol use, squamous dysplasia may occur and subsequently develop into early ESCC. Left untreated, it will develop into ESCC. For EAC after esophageal injury by gastric acid, metaplasia of nonkeratizing squamous epithelium to nonciliated, mucin-producing columnar cells occurs. Under persistent stress, the metaplasia progresses to EAC.

**Figure 2 cancers-16-00318-f002:**
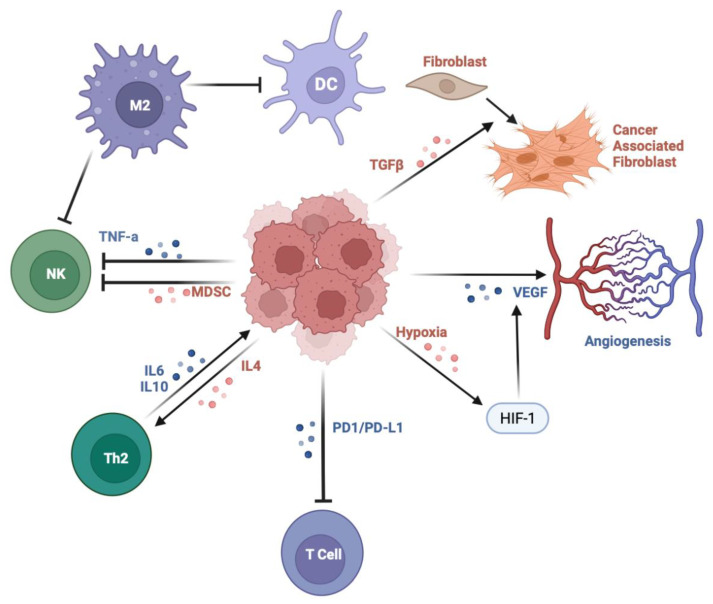
The tumor microenvironment of EC. DC: dendritic cell, NK: natural killer cell, M2: 2 macrophage, HIF-1: hypoxia-induced factor 1. The TME consists of multiple heterogeneous cells. Dendritic cells process tumor-derived antigens and are impaired from properly functioning in the TME. M2 macrophages lead to pro-survival pathway activation. Transformation of fibroblasts into cancer-associated fibroblasts leads to angiogenesis and remodeling of the extracellular matrix. HIF1 induces angiogenesis. NK cells are inhibited in the TME. The PD1/PDL-1 axis impairs T cell function.

**Figure 3 cancers-16-00318-f003:**
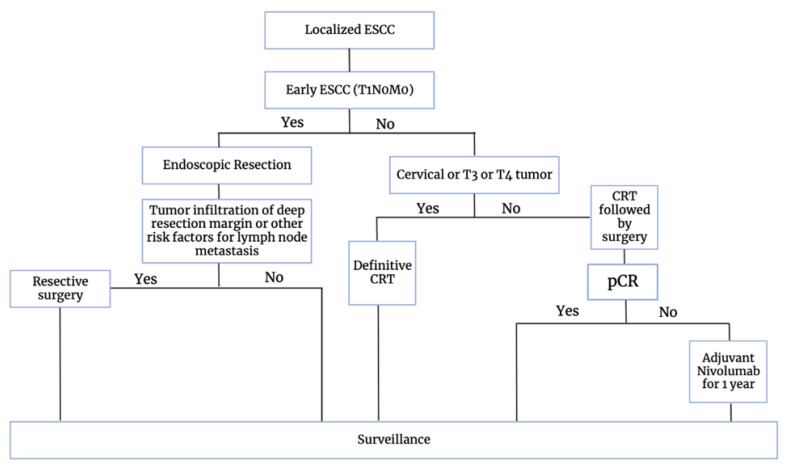
ESMO recommended guidelines for the management of ESCC.

**Table 1 cancers-16-00318-t001:** Neoadjuvant immunotherapy trials and studies.

Trial	Phase	# Patients	Pathology	Clinical Stage	Immune Checkpoint Inhibitor	Immune Target	Chemotherapeutic Agents	Radiotherapy	Primary Endpoint	Pathologic Complete Response (pCR)	Safety-Grade ≥ 3 AE
PALACE-1 [21]	Ib	20	ESCC	II-IVA	Pembrolizumab	PD-1	Carboplatin, Paclitaxel	23 fractions of 1.8 Gy	Safety	55.60%	65%
PERFECT [22]	II	40	EAC	II-IVA	Atezolizumab	PD-L1	Carboplatin, Paclitaxel	23 fractions of 1.8 Gy	Feasibility	40%	30%
ESONICT-1 [26]	II	30	ESCC	III-IV	Sintilimab	PD-1	Cisplatin, Albumin-bound paclitaxel	N/A	pCR, AEs	21.70%	3%
ESONICT-2 [29]	II	20	ESCC	III-IVA	Toripalimab	PD-1	Cisplatin, Docetaxel	N/A	pCR, AEs	16.70%	20%
SIN-ICE [23]	Pilot Study	23	ESCC	II-IVA	Sintilimab	PD-1	Docetaxel/Albumin-bound paclitaxel, Nedaplatin	N/A	pCR, safety	35.30%	30.40%
PEN-ICE [27]	II	18	ESCC	II-IVA	Pembrolizumab	PD-1	Platinum-based two drug	N/A	Safety, Efficacy	46.20%	27.80%
TD-NICE [28]	II	45	ESCC	II-IVA	Tislelizumab	PD-1	Nab-paclitaxel, Carboplatin	N/A	Major Pathologic Response (MPR)	50%	42.20%
NIC-ESCC2019 [30]	II	56	ESCC	II-IVA	Camrelizumab	PD-1	Nab-paclitaxel, cisplatin	N/A	pCR	13.70%	10.70%
Shen et al. [24]	II	28	ESCC	II-IVA	Nivolumab, Pembrolizumab, Camrelizumab	PD-1	Nab-paclitaxel, Carboplatin	N/A	Safety, Feasibility	40.70%	7.10%
Yang et al. [25]	Pilot	16	ESCC	II-IVA	Camrelizumab	PD-1	Paclitaxel, Carboplatin	N/A	pCR	31.30%	N/A (only mild and tolerable AE)
Xing et al. [31]	II	30	ESCC	II-IVA	Toripalimab	PD-1	Paclitaxel, Cisplatin	N/A	pCR	36%	6.67%
Yang et al. [32]	Pilot	23	ESCC	II-III	Camrelizumab	PD-1	Nab-paclitaxel, Carboplatin	N/A	Safety, Feasibility	25%	47.80%
He et al. [20]	II	20	ESCC	III-IVA	Toripalimab	PD-1	Paclitaxel, Carboplatin	N/A	Safety, Feasibility, MPR	18.80%	20%
Liu et al. [33]	II	60	ESCC	III-IVA	Camrelizumab	PD-1	Nab-paclitaxel, Carboplatin	N/A	pCR	39.20%	56.70%
Wang et al. [34]	Ib	30	ESCC	II-III	Camrelizumab	PD-1	Nab-paclitaxel, nedaplatin, apatinib	N/A	Safety	24.10%	36.70%

## Data Availability

Data is contained within article. No data analysis was conducted for manuscript.

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
