# Peer review of "Advancing Esophageal Cancer Treatment: Immunotherapy in Neoadjuvant and Adjuvant Settings"

_cancers, 2024, doi:10.3390/cancers16020318_

Round 1

Reviewer 1 Report

Comments and Suggestions for Authors

Very well written and comprehensive.

My only suggestions for improvement would be (1) a discussion of ongoing or future trials in the areas of neoadjuvant and adjuvant immunotherapy for esopahgeal/GEJ cancer, (2) better integration of the biomarkers section into concrete recommendations for current use or future studies, (3) a broader consideration of immunotherapy options in the "future directions" section-- CAR-T and vaccine strategies are but two of many parallel lines of investigation (for example, TIL strategies, adjuvants or intra-tumoral strategies, next-generation checkpoint inhibition etc).

Comments on the Quality of English Language

Minor edits and proofreading suggested. There are some bold paragraphs that need to be formatted to fit with the rest of the manuscript in the "biomarkers" section.

Author Response

We appreciate the constructive feedback and comments provided by Reviewer 1. Following thorough discussions, we have made extensive modifications to the manuscript in response to their valuable input.

(1) 2 Supplemental Tables have been added detailing ongoing phase 3 clinical trials in the areas of neoadjuvant and adjuvant immunotherapy for esophageal cancer.

(2)Biomarkers section has been better integrated into the sections on current use and future directions.

(3) Significant updates to the future directions sections now including "next generation checkpoint inhibitors & tumor infiltrating lymphocyte therapy"

Reviewer 2 Report

Comments and Suggestions for Authors

The present Review should be thoroughly updated in the light of the data presented in ESMO 2023

Author Response

Thank you to the reviewer for the comments.

A new section has been added to the paper "Current Treatment Strategies-ESMO 2023". A treatment algorithm adapted from ESMO is now incorporated as well.

Reviewer 3 Report

Comments and Suggestions for Authors

As a review paper on LAEC, the increase in treatment efficacy due to the use of adjustments was explained. A well-organized review paper, but I'd like you to supplement the following.

These days, complementary and alternative medicine is gradually developing and is widely used for cancer treatment. As an adjuvant for cancer treatment, I wonder if there are any known traditional medicine and if there are any known ones, I hope you can organize them.

In addition, the author mentioned that inhibitors that inhibit specific proteins of various immune cells are used as adjuvants. Various ion channels are known to play various roles in the immune cell membrane. Please explain the relationship between adjuvants and ion channels.

Author Response

The response from the reviewer is greatly appreciated.

Response:

Though there are studies on the use of alternative and complementary medicine in EC, with some studies showing improved symptoms associated with treatment with surgery and chemoradiation after the use of traditional medicine, this is beyond the scope of our study which focuses largely on immunotherapy as a complement/ alternative to current therapies (Cao). The study of traditional or alternative medicine is an interesting topic for future study.

Cao L, Wang X, Zhu G, et al. Traditional Chinese Medicine Therapy for Esophageal Cancer: A Literature Review. Integr Cancer Ther. 2021;20:15347354211061720. doi:10.1177/15347354211061720 

Response:

Studies have demonstrated the significant role of ion channels in the development and pathogenesis of upper GI cancers including esophageal cancer (specifically esophageal squamous cell carcinoma). The relationships between cancer and ion channels or immune cell membrane/ adjuvant treatments with ion channels are beyond the scope of this paper. The ion channels are currently not the focus of trials on immunotherapy for esophageal cancer, as the main targets of immunotherapy are three: PD-1, PD-L1, and CTLA-4 without specific consideration of ion channels. The role of ion channels in adjuvant therapy, or any form of immunotherapy, would also be a fascinating topic for future study.

Shiozaki A, Marunaka Y, Otsuji E. Roles of Ion and Water Channels in the Cell Death and Survival of Upper Gastrointestinal Tract Cancers. Front Cell Dev Biol. 2021;9:616933. Published 2021 Mar 11. doi:10.3389/fcell.2021.616933

Reviewer 4 Report

Comments and Suggestions for Authors

The authors have prepared a good review on neoadjuvant and adjuvant immunotherapy in the context of the treatment of locally advanced esophageal cancer. I think it will be useful and in demand by both practicing oncologists and scientists involved in research in this and related fields. Meanwhile, authors should pay special attention to a number of the following points:

1. Since there are a significant number of reviews in this area, the introduction should clearly delineate the area that this review covers and highlight the novelty of this review.

2. In addition, to strengthen the practical component of this review, I strongly recommend that the authors, in conclusion, based on systematized data, propose an algorithm for choosing a treatment regimen.

Author Response

Thank you to the reviewer for their comments. Both comments have been incorporated into the manuscript.

  1. Section added to introduction : "This manuscript focuses on the role of immunotherapy in neoadjuvant and adjuvant settings for LAEC. It identifies significant pathways in the tumor microenvironment (TME) and other biomarkers that advocate for the use in EC, while also discussing novel targeted therapies."

2. ESMO 2023 has a guidance paper on the treatment strategies for ESCC and EAC which have been incorporated into the paper. A treatment algorithm figure has also been created.

Reviewer 5 Report

Comments and Suggestions for Authors

The manuscript entitled ‘Advancing Esophageal Cancer Treatment: Immunotherapy in Neoadjuvant and Adjuvant Settings’ by Daneil Park and colleagues was well received. The article focuses on covering available literature on management of LAEC in adjuvant and neo-adjuvant settings in the context of immunotherapy. Here are some suggestions

In the abstract section font size/style is not consistent

A detailed description of Figure-1 and Figure-2 should be provided.

The overall formatting of the paper need improvement and unifications.

Lines 265-280 are bold., and it looks like a copy paste text. Please address this issue

Author Response

Thank you to the reviewer for their comments. All the suggestions have been incorporated.

  1. Abstract section font size/style is now consistent
  2. A detailed description of Figures 1 & 2 are provided 
  3. Formatting changes have been made
  4. 265-280 have been fixed

Round 2

Reviewer 3 Report

Comments and Suggestions for Authors

It is well revised.